# OpenReview forum: "Towards Safe Concept Transfer of Multi-Modal Diffusion via Causal Representation Editing"
_NeurIPS.cc/2024/Conference — NeurIPS 2024 poster_

### Official Review · Reviewer_Qo9h · 2024-07-07

**Soundness:** 4
**Presentation:** 4
**Contribution:** 4
**Rating:** 7
**Confidence:** 3

**Summary:**

The paper addresses the potential misuse of VL2I diffusion models, such as copying artistic styles without permission, which could lead to legal and social issues. The paper introduces an early exploration of safe concept transfer in MLLM-enabled diffusion models using a novel framework called Causal Representation Editing (CRE). CRE allows for effective inference-time removal of unsafe concepts from noisy images while retaining other generated content. This is achieved through fine-grained editing based on identifying the causal period during which unsafe concepts appear. The approach reduces the editing overhead by nearly half compared to existing methods. The paper presents extensive evaluations demonstrating that CRE outperforms existing methods in terms of effectiveness, precision, and scalability, even in complex scenarios with incomplete or blurred features of unsafe concepts.

**Strengths:**

1. The paper introduces an innovative approach to address the emerging concern of safe concept transfer in multimodal diffusion models. The originality is evident in its novel application of CRE to selectively remove unsafe concepts from generated images.
2. The research is methodologically sound, with comprehensive evaluations and rigorous testing of the proposed CRE framework.
3. The paper is well-written and clearly structured, making it accessible to both experts and those new to the field.
4. The significance of this work lies in its potential to influence the future development of safe AI-generated content. As AI models become increasingly integrated into creative industries, the ability to safely and efficiently remove unsafe or unwanted concepts is critical.

**Weaknesses:**

1. The paper lacks a user study or feedback from practitioners who might use the CRE framework in their workflows.
2. Although the paper provides extensive evaluations, it could benefit from a more diverse set of evaluation metrics and benchmarks. The effectiveness, precision, and scalability are well-documented, but additional metrics such as computational efficiency, user satisfaction, or real-world applicability could provide a more comprehensive assessment of the method's performance. Furthermore, comparisons with a wider range of existing techniques would strengthen the argument for the superiority of the proposed approach.
3. While the paper highlights the importance of safe content generation, it could delve deeper into the ethical and legal implications of using CRE. For example, the criteria for defining "unsafe concepts" are not fully explored, which could lead to subjective or inconsistent applications of the method. Moreover, the potential misuse of the technology for censorship or manipulation of content raises important ethical concerns that the paper does not adequately address. A more detailed discussion on these aspects would enhance the comprehensiveness of the research.

**Questions:**

Please see weaknesses.

**Limitations:**

The authors have adequately described the limitations and potential negative societal impact of their work.

---

> ### Author Rebuttal · Authors · 2024-08-07
>
> Thank you for your valuable feedback! Below we address your concerns. Please feel free to post additional comments if you have further questions.
>
> 1. Thanks for your suggestion.  We are applying our technology to more actual deployed generative models and considering inviting users to participate in testing.
> For computational efficiency, we report the comparison of inference time on generating 100 images as follows:
>
> |  Kosmos-G   | SLD  | ProtoRe | CRE |
> |  ----  | ----  |  ----  | ----  |
> |  226 s  | 228 s   | 257 s  | 246 s |
>
> We also compare with two unlearning-based methods CA [1] and UCE [2]. Experimental results are shown in anonymized pdf.
>  https://anonymous.4open.science/r/Exp-E7E7/ If an error is displayed online, please download the pdf file.
>
> [1] Kumari, Nupur, et al. "Ablating concepts in text-to-image diffusion models." Proceedings of the IEEE/CVF International Conference on Computer Vision. 2023.
>
> [2] Gandikota, Rohit, et al. "Unified concept editing in diffusion models." Proceedings of the IEEE/CVF Winter Conference on Applications of Computer Vision. 2024.
>
> 2. The concept of "unsafe content" is not yet fully defined and may require further clarification in line with laws and regulations. Our approach is designed for generation service providers to prevent the creation of such content within their models. However, there remains a potential risk of misuse of representation editing techniques. For instance, adversaries could exploit this technology to conceal a model's ability to generate specific unsafe concepts, thereby evading third-party platform (such as Hugging face) reviews. Additionally, they could intentionally introduce or remove certain concepts in the images provided by regular users, leading to biased generation outcomes.

---

> > ### Comment · Reviewer_Qo9h · 2024-08-11
> >
> > I acknowledge having read the authors' rebuttal. My overall assessment of the paper remains unchanged, and I continue to support my current rating.

---

### Official Review · Reviewer_33ja · 2024-07-12

**Soundness:** 3
**Presentation:** 3
**Contribution:** 3
**Rating:** 5
**Confidence:** 3

**Summary:**

The authors studied an important problem about misuse of Text-to-image (T2I) diffusion model, leading to legal and social issues. They propose a causal representation editing (CRE) method, extends representation editing from large language models to diffusion-based models. CRE improves safe content generation by intervening at diffusion timesteps linked to unsafe concepts, allowing precise removal of harmful content while preserving quality. The extensive experiment results have shown the effectiveness of their model.

**Strengths:**

1. The authors focus on an important problem about misuse of Text-to-image (T2I) diffusion model, leading to legal and social issues.
2. The writing is clear and easy to follow.
3. The authors conduct extensive experiments to verify the effectiveness of their proposed method, CRE.

**Weaknesses:**

1. The authors may need the experiment to compare the inference time with other Inference-time Safe Concept Transfer models to further verify the efficiency of their model.
2. It’s better to provide more related works about representation editing in the related works.
3. The contribution of the design of selecting editing timesteps to performance of the model is unclear compared with random selection.

**Questions:**

In ProtoRe, they also have similar editing function in their Eq (7). The contribution in this paper is about adding the term related to discriminator. Can authors explain more about the contribution compared with ProtoRe?

**Limitations:**

Can be found in weaknesses part.

---

> ### Author Rebuttal · Authors · 2024-08-07
>
> Thank you for your valuable feedback! Below we address your questions and concerns. Please feel free to post additional comments if you have further questions.
>
> **For weakness 1:** We report the comparison of inference time on generating 100 images as follows:
> |  Kosmos-G   | SLD  | ProtoRe | CRE |
> |  ----  | ----  |  ----  | ----  |
> |  226 s  | 228 s   | 257 s  | 246 s |
>
> **For weakness 2:** Thanks for your suggestion. We include more related works to comprehensively introduce the advancements of representation editing for LLMs.
>
> Representation editing involves creating steering vectors that, when added during the forward passes of a frozen large language model (LLM), produce desired changes in the output text [1,2]. It is based on the idea that LLMs encode knowledge linearly [3]. By editing these steering vectors, which are derived from the model's activations, users can modify the model's behavior [4]. Some previous studies have used gradient descent to search these steering vectors [5,6]. Current studies on Inference-Time Intervention (ITI) [7]  in Large Language Models (LLMs) indicate that many LLMs exhibit interpretable directions in their activation spaces, which influence their inference processes. For instance, by introducing carefully designed steering vectors to specific layers for particular tokens, the output can be significantly biased, regardless of the user prompt's topic [8,9]. Developing a training-free editing method to mitigate unsafe concepts in generative models offers two key advantages. Firstly, it allows the model to retain its strong zero-shot generation ability by preserving the knowledge from pre-training. Secondly, as unsafe concepts may change dynamically due to legal or copyright factors, a plug-and-play editing method can efficiently add or remove types of unsafe concepts under governance.
>
> [1] Dathathri, Sumanth, et al. "Plug and play language models: A simple approach to controlled text generation." arXiv preprint arXiv:1912.02164 (2019).
>
> [2] Zou, Andy, et al. "Representation engineering: A top-down approach to ai transparency." arXiv preprint arXiv:2310.01405 (2023).
>
> [3] Burns, Collin, et al. "Discovering latent knowledge in language models without supervision." arXiv preprint arXiv:2212.03827 (2022).
>
> [4] Li, Kenneth, et al. "Emergent world representations: Exploring a sequence model trained on a synthetic task." arXiv preprint arXiv:2210.13382 (2022).
>
> [5] Subramani, Nishant, Nivedita Suresh, and Matthew E. Peters. "Extracting latent steering vectors from pretrained language models." arXiv preprint arXiv:2205.05124 (2022).
>
> [6] Hernandez, Evan, Belinda Z. Li, and Jacob Andreas. "Inspecting and editing knowledge representations in language models." arXiv preprint arXiv:2304.00740 (2023).
>
> [7] Li, Kenneth, et al. "Inference-time intervention: Eliciting truthful answers from a language model." Advances in Neural Information Processing Systems 36 (2024).
>
> [8] Turner, Alex, et al. "Activation addition: Steering language models without optimization." arXiv preprint arXiv:2308.10248 (2023).
>
> [9] Liu, Sheng, Lei Xing, and James Zou. "In-context vectors: Making in context learning more effective and controllable through latent space steering." arXiv preprint arXiv:2311.06668 (2023).
>
> **For weakness 3:** We conduct a comparison of CRE and random selection. For fairness, the total number of edits in random selection is similar as that in CRE. Experimental results are shown in Figure 5 in the anonymized pdf.
> https://anonymous.4open.science/r/Exp-E7E7/ If an error is displayed online, please download the pdf file.
>
> **For question:** Given the limitations of ProtoRe, our proposed CRE method introduces improvements in two key areas:
>
> **Scalability:** ProtoRe struggles with scalability; its performance deteriorates as the number of unsafe concepts increases, particularly when editing multiple concepts simultaneously. To address this, CRE incorporates a discriminator that focuses on editing the representation of only one unsafe concept at a time. This approach helps maintain stable performance.
>
> **Efficiency:** ProtoRe applies representation editing throughout the entire diffusion process, leading to unnecessary computational overhead. In contrast, CRE targets specific diffusion steps associated with unsafe concepts, based on an understanding of the different information generated in successive diffusion process. Typically, CRE restricts editing to no more than half of the diffusion steps, thereby reducing overhead.

---

### Official Review · Reviewer_MP63 · 2024-07-13

**Soundness:** 2
**Presentation:** 3
**Contribution:** 2
**Rating:** 6
**Confidence:** 3

**Summary:**

The paper introduces Causal Representation Editing (CRE) for vision-language-to-image (VL2I) models to prevent undesirable concept generation. CRE prevents unsafe image generation through the following process:
1. Using a discriminator, it detects whether an unsafe concept is present in the user prompt.
2. If the discriminator identifies an unsafe concept, it projects out the representation $\tilde{A}$ corresponding to the unsafe concept from the existing attention representation $A$.
3. The projection out is not performed at every diffusion timestep but within the specific interval $[t_s, t_e]$ identified as causally influential for unsafe concept generation through assess-with-exclusion.
The authors demonstrated the efficacy of CRE through object, style, and multiple-style censoring.

**Strengths:**

(Here, I will refer to "preventing unsafe image generation" as censoring.)

1. Ability to Handle Unsafe Image Prompts
Most existing safety methods are designed for T2I models and focus on preventing the generation of unsafe text concepts. The method proposed in this paper has the advantage of being applicable regardless of the prompt domain, whether text or image.

2. Censoring Performance
Table 1 shows that CRE performs better in censoring compared to existing methods.

**Weaknesses:**

1. Using an external discriminator is highly inefficient.

For example, let's say we need to remove K unsafe concepts. Using an external discriminator adds two types of overhead: (1) training a discriminator for each concept and (2) performing discriminator inference for each concept. While (1) can be somewhat mitigated with models like CLIP that have good zero-shot classification performance, (2) remains problematic. In my opinion, the advantage of test-time guidance methods like ActAdd and SLD is that they don't require training. The use of a discriminator diminishes the benefits they offer over inference-time refusal or machine unlearning methods.

2. Inadequate experiments

The current paper lacks significant experiments needed in the safety unlearning field.

There are no experiments showing the impact of CRE on image fidelity. While a perfect discriminator would be ideal, real-world discriminators are not perfect. Can you report the COCO 30k dataset FID? Considering limited computational resources, knowing the results when censoring "cassette player" and "Disney" would be sufficient.

This paper uses a T2I diffusion model as an image decoder, so it could compare machine unlearning techniques for T2I diffusion models as baselines. Are there any comparisons with safety unlearning techniques for T2I diffusion models? State-of-the-art methods include SPM [1] and MACE [2], but considering the submission time, comparing with CA [3] and UCE [4] would suffice.

There are no experiments on NSFW concept removal. I'm curious if CRE can robustly prevent adversarial prompt attacks like Ring-A-Bell [5] or concept inversion [6] through red teaming.

3. Methodological novelty (minor)
Directly using the representation editing method ActAdd from LLMs is relatively less novel.

---

[1] : One-Dimensional Adapter to Rule Them All: Concepts, Diffusion Models and Erasing Applications, https://arxiv.org/abs/2312.16145

[2] : MACE: Mass Concept Erasure in Diffusion Models, https://arxiv.org/abs/2403.06135

[3] : Ablating Concepts in Text-to-Image Diffusion Models, https://arxiv.org/abs/2303.13516

[4] : Unified Concept Editing in Diffusion Models, https://arxiv.org/abs/2308.14761

[5] : Ring-A-Bell! How Reliable are Concept Removal Methods for Diffusion Models?, https://arxiv.org/abs/2310.10012

[6] : Circumventing Concept Erasure Methods For Text-to-Image Generative Models, https://arxiv.org/abs/2308.01508

**Questions:**

Q1: Can you provide the specific values of $[t_s, t_e]$ obtained through assess-with-exclusion? The current paper includes the method but lacks actual causal tracing results or precise values for each concept.

**Limitations:**

The author describes the limitations caused by discriminator.

---

> ### Author Rebuttal · Authors · 2024-08-07
>
> Thank you for your valuable feedback! Below we address your questions and concerns. Please feel free to post additional comments if you have further questions.
>
> **For weakness 1:** We emphasize the necessity of the discriminator by illustrating the disadvantages of existing inference-time refusal methods. Taking ProtoRe as an example, ProtoRe uses CLIP to cluster unsafe features, which is efficient but not scalable. As the number of unsafe concepts increases, the absence of a discriminator necessitates editing the representations of multiple unsafe concepts simultaneously, even during safe content generation, which significantly degrades image quality. The experimental results presented in Table 3 and Figure 3 demonstrate that the effectiveness of simultaneous representation editing for multiple concepts is limited when conducted without the assistance of a discriminator. To solve this problem, we introduce a discriminator to perform representation editing on only a single unsafe concept at a time, ensuring stable performance.
>
> Compared with unlearning, CRE has two outstanding advantages: 1. It does not require full fine-tuning of the generated model; 2. It can flexibly add and delete the types of unsafe concepts currently supervised.
>
> **For weakness 2:** Thanks for your suggestion. We report the COCO 30k dataset FID of the model after introducing CRE for cassette player and Mickey Mouse (see Figure 2 in the anonymized pdf ):
> https://anonymous.4open.science/r/Exp-E7E7/
> If an error is displayed online, please download the pdf file.
>
> | Model |  Kosmos-G   | Kosmos-G w. CRE(cassette player)  | Kosmos-G w. CRE(Mickey Mouse) |
> |  ----  | ----  |  ----  | ----  |
> | FID |  10.99 | 13.83 | 11.34  |
>
> It is important to note that CRE is only activated when the discriminator detects that the input prompt contains either concept cassette player or Mickey Mouse. CRE does not alter the reasoning process for regular users, and thus does not impact the quality of image generation for them.
>
> We compare with CA and UCE. Experimental results of CA are shown in Figure 3 in the anonymized pdf.
>
> | ImageNet Category |  Cassette Player   | Chain Saw | Church | Gas Pump | Tench | Garbage Truck| English Springer | Golf Ball |Parachute | French Horn | Avg |
> |  ----  | ----  |  ----  | ----  |  ----  | ----  |  ----  | ----  | ----  | ----  | ----  | ----  |
> | UCE |  0.0 | 0.0 | 8.4  | 0.0 | 0.0 | 14.8 | 0.2 | 0.8 |  1.4| 0.0 | 2.6 |
> | CRE |  0.0 | 0.0 | 0.0  | 0.0 | 0.0 | 0.0 | 0.0 | 0.0 |   0.0 | 0.0 | 0.0 |
>
> We conduct experiment on NSFW Inappropriate Image Prompts (I2P) benchmark dataset, which contains 4703 toxic prompts assigned to at least one of the following categories: hate, harassment, violence, self-harm, sexual, shocking, illegal activity. See Figure 1 in the anonymized pdf for examples. Numerical results are as follows:
>
> | I2P Category |  Hate   | Harrassment  | Violence | Self-harm | Sexual | Shocking | Illegal activity | Avg |
> |  ----  | ----  |  ----  | ----  |  ----  | ----  |  ----  | ----  | ----  |
> | SLD |  0.2 | 0.17 | 0.23  | 0.16 | 0.14 | 0.30 | 0.14 | 0.19 |
> | ProtoRe |  0.1 | 0.07 | 0.09  | 0.09 | 0.08 | 0.1 | 0.11 | 0.09 |
> | CRE |  0.04 | 0.07 | 0.07  | 0.06 | 0.07 | 0.06 | 0.04 | 0.06 |
>
> Due to time limitation, the defense experiments against adversarial prompt attacks, such as Ring-A-Bell and concept inversion, will be included in a later version of the paper.
>
> **For weakness 3:** Comparing with directly representation editing (ActAdd and ProtoRe), CRE makes improvements in two aspects.
> First, we introduced the discriminator to solve the scalability problem (as described above).
> Second, ProtoRe performs representation editing throughout the diffusion process, causing unnecessary overhead. Based on the understanding of the different dimensional information generated at each stage of the diffusion process, we only edit in the diffusion steps related to specific unsafe concepts. In most cases, CRE only edits in no more than half of the steps.
>
> **For question Q1:** We show the editing intervals corresponding to four concepts in Figure 4 in the anonymized pdf.

---

> > ### Comment · Area_Chair_sAnN · 2024-08-08
> >
> > According to the Author's Guidelines (sent by email),
> > > 4. All the texts you post (rebuttal, discussion and PDF) should not contain any links to external pages.
> >
> > This accident will be redirected to SAC.

---

> > > ### Author Response · Authors · 2024-08-08
> > >
> > > We apologize for any inconvenience caused. Our aim is to address the reviewer's concerns as thoroughly as possible within the constraints of a double-blind review. The anonymous link provided contains only an anonymized PDF, ensuring no personal information is disclosed. The images included in the PDF are supplementary experiments intended to address the reviewer's concerns. We regret any adverse effects this may have caused.

---

> > > > ### Comment · Reviewer_MP63 · 2024-08-09
> > > >
> > > > I hope that the minor mistake of uploading the link does not significantly impact the evaluation of this research. I appreciate your sincere efforts.
> > > >
> > > > I had three major concerns:
> > > >
> > > > > The use of the discriminator seems inefficient.
> > > >
> > > > In my view, it still appears inefficient to use the discriminator in the representation. However, after comparing the baseline performance below, I now find this level of inefficiency to be understandable.
> > > >
> > > > > I was concerned that it might excessively degrade the model's capability.
> > > >
> > > > Regarding the rebuttal on FID and 4T2N, it does seem to impact adherence to the text description compared to existing methods. Nevertheless, this concern has been alleviated with the improvement of the baseline.
> > > >
> > > > > Missing baseline (especially the unlearning baseline).
> > > >
> > > > While UCE is not state-of-the-art, the results are convincing enough for me. This research holds significant value, particularly because it offers strengths in safety compared to unlearning methods.
> > > >
> > > > Overall, my concerns have been partially addressed. Therefore, I will raise the score to 6.

---

> ### Comment · Area_Chair_sAnN · 2024-08-08
>
> Reviewers, please refrain from opening the link. I will inform you of any updates.
>
> In the meantime, feel free to continue the discussion as usual.

---

### Official Review · Reviewer_4T2N · 2024-07-13

**Soundness:** 3
**Presentation:** 3
**Contribution:** 3
**Rating:** 5
**Confidence:** 3

**Summary:**

This paper proposes a framework called Causal Representation Editing (CRE) to address the ethical and copyright concerns in vision-language-to-image (VL2I) diffusion models. CRE enhances safe content generation by intervening at diffusion timesteps linked to unsafe concepts, effectively removing harmful content while preserving quality. The approach is more effective, precise, and scalable than existing methods. CRE also offers a solution for complex scenarios, providing insights into managing harmful content in diffusion-based models. Comprehensive evaluations demonstrate CRE's superiority in various benchmarks.

**Strengths:**

1. Innovative Causal Representation Editing Framework: The paper introduces a novel framework called Causal Representation Editing (CRE) that effectively extends representation editing techniques from language models to diffusion-based generative models. This framework enhances the efficiency and flexibility of safe content generation, providing a new approach to addressing ethical and copyright concerns in vision-language-to-image (VL2I) models.

2. Comprehensive Handling of Unsafe Concepts: CRE demonstrates superior effectiveness, precision, and scalability compared to existing methods. It can handle complex scenarios, including incomplete or blurred representations of unsafe concepts, ensuring that harmful content is precisely removed while maintaining acceptable content quality.

3. Detailed Experimental Validation: The paper provides extensive evaluations and experiments to validate the effectiveness of the proposed method. The results show that CRE surpasses existing methods in various benchmarks, highlighting its potential for managing harmful content generation in diffusion-based models.

**Weaknesses:**

1. Dependency on Discriminator Accuracy: The effectiveness of CRE is heavily reliant on the accuracy of the unsafe concept discriminator. If the discriminator fails to accurately identify unsafe concepts, CRE might incorrectly edit safe content, leading to unnecessary modifications and potentially impacting the quality of the generated images.

2. Additional Inference Overhead: Compared to safe generation methods that use fine-tuned diffusion models, CRE introduces additional inference overhead. This can lead to increased computation time and resource usage, which might be a significant drawback in practical applications where efficiency is crucial.

3. Limited Applicability to Complex Unsafe Concepts: While CRE is effective for well-defined unsafe concepts, its performance might degrade when dealing with more complex or nuanced unsafe concepts that are difficult to categorize or describe. This limitation restricts its applicability in real-world scenarios where unsafe content is not easily defined.

**Questions:**

1. Could you explain how your method handles cases where the discriminator accuracy is low, and CRE might perform representation editing even for safe prompts? Are there any mechanisms to mitigate this issue?

2. The paper mentions that the additional overhead introduced by representation editing is within a tolerable range. Could you provide more details on how this overhead might scale with larger datasets or more complex models?

**Limitations:**

see weaknesses

---

> ### Author Rebuttal · Authors · 2024-08-07
>
> Thank you for your valuable feedback! Below we address your questions and concerns. Please feel free to post additional comments if you have further questions.
>
> **For Weakness 1 and Question 1:**
> As the reviewer mentioned, the discriminator may not always make accurate judgments. To assess the impact of our CRE on safe generation when the classifier fails, we conducted an experiment using ImageNet, as shown in Table 1. Specifically, we examined the effect of consistently applying representation editing to other safe categories. For instance, we introduced CRE for the category "cassette player" and then measured the average image generation accuracy for the remaining nine categories. The results are as follows:
>
> | Model  | Kosmos-G | Kosmos-G w. ProtoRe (cassette player)  | Kosmos-G w. CRE(cassette player)  |
> |  ----  | ----  |  ----  | ----  |
> | Avg Acc |  40.44 | 24.69 | 33.84 |
>
> When the discriminator incorrectly classifies a safe concept as unsafe, the generation of safe content is only minimally impacted. Additionally, we report the FID score on the COCO 30k dataset after applying CRE to the "cassette player" and "Mickey Mouse".
>
> | Model |  Kosmos-G   | Kosmos-G w. CRE(cassette player)  | Kosmos-G w. CRE(Mickey Mouse) |
> |  ----  | ----  |  ----  | ----  |
> | FID |  10.99 | 13.83 | 11.34  |
>
> While occasional errors by the discriminator do not significantly affect safe content generation, we emphasize its importance for scalability. As the number of unsafe concepts increases, the absence of a discriminator necessitates editing the representations of multiple unsafe concepts simultaneously, even during safe content generation, which significantly degrades image quality. The experimental results presented in Table 3 and Figure 3 demonstrate that the effectiveness of simultaneous representation editing for multiple concepts is limited when conducted without the assistance of a discriminator.
>
> **For Weakness 2 and Question 3:**
> As demonstrated, increasing the number of unsafe concepts gradually decreases the accuracy of safe content generation. Therefore, the discriminator effectively prevents image quality loss by avoiding simultaneous editing of multiple concepts. We shift the cost of managing additional unsafe concepts from simultaneous editing during inference to the pre-training phase of the discriminator. The inference cost of CRE remains comparable to that of the standard model. We report the comparison of inference time on generating 100 images as follows:
>
> |  Kosmos-G   | SLD  | ProtoRe | CRE |
> |  ----  | ----  |  ----  | ----  |
> |  226 s  | 228 s   | 257 s  | 246 s |
>
> It is important to note that CRE can be easily extended to larger models, even as the model structure becomes more complex and the number of parameters increases. This is because CRE only edits the intermediate outputs of the cross-attention layer in the diffusion model, and the parameters in this layer account for just **5.11%** of the total model parameters.
>
> **For Weakness 3:**
> The effectiveness of representation editing depends on aligning the encoding space of unsafe concepts with the feature space of the generative model, which can be achieved using a general encoder like CLIP. However, there is a limitation: if unsafe concepts are not clearly defined, they cannot be correctly encoded by the encoder. We will discuss this limitation in the article.

---

### Author Response · Authors · 2024-08-08

We apologize for including the anonymous link in the rebuttal process. We confirm that the PDF contained in the link does not disclose any personal information and adheres to the double-blind review mechanism. The PDF contains supplementary experimental images addressing the reviewer's concerns. If this has caused any adverse effects, we sincerely apologize and request the Area Chair to remove the link.

---

> ### Comment · Area_Chair_sAnN · 2024-08-10
>
> The accident has been reported to SAC, but they have not responded yet. We hope reviewers refrain from opening the link and are not affected by it in their assessments.
>
> Unless SAC's explicit action, the author-reviewer discussion will continue.

---

> > ### Author Response · Authors · 2024-08-10
> >
> > We have deleted the anonymous PDF. We sincerely apologize for this oversight and extend our gratitude to all the reviewers and area chairs for their efforts.

---

### Decision · Program_Chairs · 2024-09-25

**Decision:**

Accept (poster)

**Comment:**

Current approaches to address machine safety, including dataset filtering and machine unlearning, often lack scalability or effectiveness. The proposed framework, causal representation editing (CRE), argues that it offers a more effective and scalable solution by targeting unsafe content during diffusion timesteps, ensuring safe content generation while maintaining quality.

The reviewers think its strengths may include:

- Innovative causal representation editing (4T2N, Qo9h)
- while having excellent performance (4T2N, MP63)
- Detailed experimental validation (4T2N, 33ja)
- Well-written and clear (33ja, Qo9h)

Initially, its weaknesses were:
- A heavy reliance on the unsafe concept discriminator (4T2N) and inefficiency (MP63). However, the rebuttal appears to have successfully addressed these concerns by providing experimental evidence.

Overall, the author rebuttals have garnered positive feedback. After reviewing the rebuttal threads, it appears that the major issues have been adequately addressed, leading to a recommendation for the paper’s acceptance.